# SEARNN:
# TRAINING RNNS WITH GLOBAL-LOCAL LOSSES

**Rémi Leblond**[1,2]*    **Jean-Baptiste Alayrac**[1,2]*    **Anton Osokin**[1,2,3]    **Simon Lacoste-Julien**[4,5]

[1]Département d'informatique de l'ENS, Paris, France
[2]INRIA, École normale supérieure, CNRS, PSL Research University
[3]National Research University Higher School of Economics, Moscow, Russia
[4]Université de Montréal & Montreal Institute for Learning Algorithms (MILA)
[5]Canadian Institute for Advanced Research (CIFAR)
`{firstname.lastname}@inria.fr`

## ABSTRACT

We propose SEARNN, a novel training algorithm for recurrent neural networks (RNNs) inspired by the "learning to search" (L2S) approach to structured prediction. RNNs have been widely successful in structured prediction applications such as machine translation or parsing, and are commonly trained using maximum likelihood estimation (MLE). Unfortunately, this training loss is not always an appropriate surrogate for the test error: by only maximizing the ground truth probability, it fails to exploit the wealth of information offered by structured losses. Further, it introduces discrepancies between training and predicting (such as exposure bias) that may hurt test performance. Instead, SEARNN leverages test-alike search space exploration to introduce global-local losses that are closer to the test error. We first demonstrate improved performance over MLE on two different tasks: OCR and spelling correction. Then, we propose a subsampling strategy to enable SEARNN to scale to large vocabulary sizes. This allows us to validate the benefits of our approach on a machine translation task.

## 1 INTRODUCTION

Recurrent neural networks (RNNs) have been quite successful in structured prediction applications such as machine translation (Sutskever et al., 2014), parsing (Ballesteros et al., 2016) or caption generation (Vinyals et al., 2015). These models use the same repeated *cell* (or unit) to output a sequence of tokens one by one. As each prediction takes into account all previous predictions, this cell learns to output the next token conditioned on the previous ones. The standard training loss for RNNs is derived from maximum likelihood estimation (MLE): we consider that the cell outputs a probability distribution at each step in the sequence, and we seek to maximize the probability of the ground truth.

Unfortunately, this training loss is not a particularly close surrogate to the various test errors we want to minimize. A striking example of discrepancy is that the MLE loss is close to 0/1: it makes no distinction between candidates that are close or far away from the ground truth (with respect to the structured test error), thus failing to exploit valuable information. Another example of train/test discrepancy is called *exposure* or *exploration bias* (Ranzato et al., 2016): in traditional MLE training the cell learns the conditional probability of the next token, based on the previous ground truth tokens – this is often referred to as *teacher forcing*. However, at test time the model does not have access to the ground truth, and thus feeds its own previous predictions to its next cell for prediction instead.

Improving RNN training thus appears as a relevant endeavor, which has received much attention recently. In particular, ideas coming from reinforcement learning (RL), such as the REINFORCE and ACTOR-CRITIC algorithms (Ranzato et al., 2016; Bahdanau et al., 2017), have been adapted to derive training losses that are more closely related to the test error that we actually want to minimize.

---

*Equal contribution.

In order to address the issues of MLE training, we propose instead to use ideas from the structured prediction field, in particular from the "learning to search" (L2S) approach introduced by Daumé et al. (2009) and later refined by Ross & Bagnell (2014) and Chang et al. (2015) among others.

**Contributions.** In Section 2, we review the limitations of MLE training for RNNs in details. We also clarify some related claims made in the recent literature. In Section 3, we make explicit the strong links between RNNs and the L2S approach. In Section 4, we present SEARNN, a novel training algorithm for RNNs, using ideas from L2S to derive a *global-local* loss that is much closer to the test error than MLE. We demonstrate that this novel approach leads to significant improvements on two difficult structured prediction tasks, including a spelling correction problem recently introduced in Bahdanau et al. (2017). As this algorithm is quite costly, we investigate scaling solutions in Section 5. We explore a subsampling strategy that allows us to considerably reduce training times, while maintaining improved performance compared to MLE. We apply this new algorithm to machine translation and report significant improvements in Section 6. Finally, we contrast our novel approach to the related L2S and RL-inspired methods in Section 7.

## 2 TRADITIONAL RNN TRAINING AND ITS LIMITATIONS

RNNs are a large family of neural network models aimed at representing sequential data. To do so, they produce a sequence of states $(h_1, ..., h_T)$ by recursively applying the same transformation (or *cell*) $f$ on the sequential data: $h_t = f(h_{t-1}, y_{t-1}, x)$, with $h_0$ an initial state and $x$ an optional input.

Many possible design choices fit this framework. We focus on a subset typically used for structured prediction, where we want to model the *joint probability* of a target sequence $(y_1, \ldots, y_{T_x}) \in \mathcal{A}^{T_x}$ given an input $x$ (e.g. the *decoder* RNN in the encoder-decoder architecture (Sutskever et al., 2014; Cho et al., 2014)). Here $\mathcal{A}$ is the alphabet of output tokens and $T_x$ is the length of the output sequence associated with input $x$ (though $T_x$ may take different values, in the following we drop the dependency in $x$ and use $T$ for simplicity). To achieve this modeling, we feed $h_t$ through a projection layer (i.e. a linear classifier) to obtain a vector of scores $s_t$ over all possible tokens $a \in \mathcal{A}$, and normalize these with a softmax layer (an exponential normalizer) to obtain a distribution $o_t$ over tokens:

$$h_t = f(h_{t-1}, y_{t-1}, x); \qquad s_t = \text{proj}(h_t); \qquad o_t = \text{softmax}(s_t) \qquad \forall\, 1 \le t \le T. \quad (1)$$

The vector $o_t$ is interpreted as the predictive conditional distribution for the $t^{\text{th}}$ token given by the RNN model, i.e. $p(a|y_1, \ldots, y_{t-1}, x) := o_t(a)$ for $a \in \mathcal{A}$. Multiplying the values $o_t(y_t)$ together thus yields the joint probability of the sequence $y$ defined by the RNN (thanks to the chain rule):

$$p(y_1, ..., y_T|x) = p(y_1|x)p(y_2|y_1, x) \ldots p(y_T|y_1, ..., y_{T-1}, x) := \Pi_{t=1}^T o_t(y_t). \quad (2)$$

As pointed by Goodfellow et al. (2016), the underlying structure of these RNNs as graphical models is thus a complete graph, and there is no conditional independence assumption to simplify the difficult prediction task of computing $\arg\max_{y \in \mathcal{Y}} p(y|x)$. In practice, one typically uses either beam search to approximate this decoding, or a sequence of *greedy* predictions $\hat{y}_t := \arg\max_{a \in \mathcal{A}} p(a|\hat{y}_1, \ldots, \hat{y}_{t-1}, x)$.

If we use the "teacher forcing" regimen, where the inputs to the RNN cell are the ground truth tokens (as opposed to its own greedy predictions), we obtain the probability of each ground truth sequence according to the RNN model. We can then use MLE to derive a loss to train the RNN. One should note here that despite the fact that the individual output probabilities are at the token level, the MLE loss involves the joint probability (computed via the chain rule) and is thus at the *sequence level*.

**The limitations of MLE training.** While this maximum likelihood style of training has been very successful in various applications, it suffers from several known issues, especially for structured prediction problems. The first one is called *exposure* or *exploration bias* (Ranzato et al., 2016). During training (with teacher forcing), the model learns the probabilities of the next tokens conditioned on the ground truth. But at test time, the model does not have access to the ground truth and outputs probabilities are conditioned on its own previous predictions instead. Therefore if the predictions differ from the ground truth, the model has to continue based on an exploration path it has not seen during training, which means that it is less likely to make accurate predictions. This phenomenon, which is typical of sequential prediction tasks (Kääriäinen, 2006; Daumé et al., 2009) can lead to a compounding of errors, where mistakes in prediction accumulate and prevent good performance.

The second major issue is the discrepancy between the training loss and the various test errors associated with the tasks for which RNNs are used (e.g. edit distance, F1 score...). Of course, a single surrogate is not likely to be a good approximation for all these errors. One salient illustration of that fact is that MLE ignores the information contained in structured losses. As it only focuses on maximizing the probability of the ground truth, it does not distinguish between a prediction that is very close to the ground truth and one that is very far away. Thus, most of the information given by a structured loss is not leveraged when using this approach.

**Local vs. sequence-level.** Some recent papers (Ranzato et al., 2016; Wiseman & Rush, 2016) also point out the fact that since RNNs output next token predictions, their loss is local instead of sequence-level, contrary to the error we typically want to minimize. This claim seems to contradict the standard RNN analysis, which postulates that the underlying graphical model is the complete graph: that is, the RNN outputs the probability of the next tokens conditioned on all the previous predictions. Thanks to the chain rule, one recovers the probability of the whole sequence. Thus the maximum likelihood training loss is indeed a *sequence level* loss, even though we can decompose it in a product of local losses at each cell. However, if we assume that the RNN outputs are only conditioned on the last few predictions (instead of all previous ones), then we can indeed consider the MLE loss as local. In this setting, the underlying graphical model obeys Markovian constraints (as in maximum entropy Markov models (MEMMs)) rather than being the complete graph; this corresponds to the assumption that the information from the previous inputs is imperfectly carried through the network to the cell, preventing the model from accurately representing long-term dependencies.

Given all these limitations, exploring novel ways of training RNNs appears to be a worthy endeavor, and this field has attracted a lot of interest in the past few years. While many papers try to adapt ideas coming from the reinforcement learning literature, we instead focus in this paper on the links we can draw with structured prediction, and in particular with the L2S approach.

## 3 LINKS BETWEEN RNNS AND LEARNING TO SEARCH

The L2S approach to structured prediction was first introduced by Daumé et al. (2009). The main idea behind it is a *learning reduction* (Beygelzimer et al., 2016): transforming a complex learning problem (structured prediction) into a simpler one that we know how to solve (multiclass classification). To achieve this, Daumé et al. (2009) propose in their SEARN algorithm to train a shared local classifier to predict each token *sequentially* (conditioned on all inputs and all past decisions), thus searching greedily step by step in the big combinatorial space of structured outputs. The idea that tokens can be predicted one at a time, conditioned on their predecessors, is central to this approach.

The training procedure is iterative: at the beginning of each round, one uses the current model (or policy[1]) to build an intermediate dataset to train the shared classifier on. The specificity of this new dataset is that each new sample is accompanied by a cost vector containing one entry per token in the output vocabulary $\mathcal{A}$. To obtain these cost vectors, one starts by applying a *roll-in* policy to predict all the tokens up to $T$, thus building one trajectory (or exploration path) in the search space per sample in the initial dataset. Then, at each time step $t$, one picks arbitrarily each possible token (diverging from the roll-in trajectory) and then continues predicting to finish the modified trajectory using a *roll-out* policy. One then computes the cost of all the obtained sequences, and ends up with $T$ vectors (one per time step) of size $|\mathcal{A}|$ (the number of possible tokens) for every sample. Figure 1 describes the same process for our SEARNN algorithm (although in this case the shared classifier is an RNN).

One then extracts features from the "context" at each time step $t$ (which encompasses the full input and the previous tokens predicted up to $t$ during the roll-in).[2] Combining the cost vectors to these features yields the new intermediary dataset. The original problem is thus reduced to multi-class *cost-sensitive* classification. Once the shared classifier has been fully trained on this new dataset, the policy is updated for the next round. The algorithm is described more formally in Algorithm 2 (see Appendix A). Theoretical guarantees for various policy updating rules are provided by e.g. Daumé et al. (2009) and Chang et al. (2015).

---

[1] Note that the vocabulary used in this literature is slightly different from that of RNNs: tokens are rather referenced as actions, predictions as decisions and models as policies.

[2] This is often referred to as "search state" in the L2S literature, but we prefer calling it context to avoid confusion with the RNN hidden state.

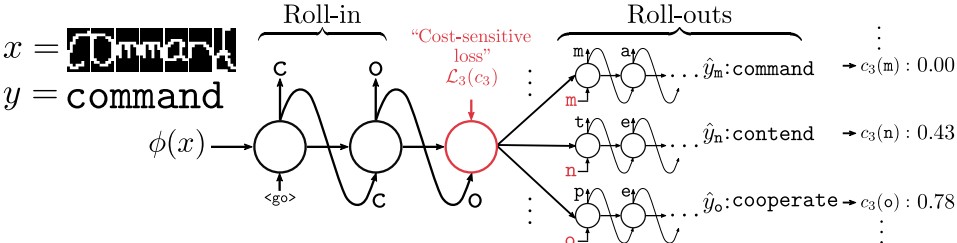

Figure 1: Illustration of the roll-in/roll-out mechanism used in SEARNN. The goal is to obtain a vector of costs for each cell of the RNN in order to define a *cost-sensitive loss* to train the network. These vectors have one entry per possible token. Here, we show how to obtain the vector of costs for the red cell. First, we use a *roll-in* policy to predict until the cell of interest. We highlight here the *learned* policy where the network passes its own prediction to the next cell. Second, we proceed to the *roll-out* phase. We feed every possible token (illustrated by the red letters) to the next cell and let the model predict the full sequence. For each token $a$, we obtain a predicted sequence $\hat{y}_a$. Comparing it to the ground truth sequence $y$ yields the associated cost $c(a)$.

**Roll-in and roll-out policies.** The policies used to create the intermediate datasets fulfill different roles. The *roll-in* policy controls what part of the search space the algorithm explores, while the *roll-out* policy determines how the cost of each token is computed. The main possibilities for both roll-in and roll-out are explored by Chang et al. (2015). The *reference* policy tries to pick the optimal token based on the ground truth. During the roll-in, it corresponds to picking the ground truth. For the roll-out phase, while it is easy to compute an optimal policy in some cases (e.g. for the Hamming loss where simply copying the ground truth is also optimal), it is often too expensive (e.g. for BLEU score). One then uses a heuristic (in our experiments the reference policy is to copy the ground truth for both roll-in and roll-out unless indicated otherwise). The *learned policy* simply uses the current model instead, and the *mixed* policy stochastically combines both. According to Chang et al. (2015), the best combination when the reference policy is poor is to use a learned roll-in and a mixed roll-out.

**Links to RNNs.** One can identify the following interesting similarities between a greedy approach to RNNs and L2S. Both models handle sequence labeling problems by outputting tokens recursively, conditioned on past decisions. Further, the RNN "cell" is shared at each time step and can thus also be seen as a shared local classifier that is used to make structured predictions, as in the L2S framework. In addition, there is a clear equivalent to the choice of roll-in policy in RNNs. Indeed, teacher forcing (conditioning the outputs on the ground truth) can be seen as the roll-in *reference policy* for the RNN. Instead, if one conditions the outputs on the previous predictions of the model, then we obtain a roll-in *learned policy*.

Despite these connections, many differences remain. Amongst them, the fact that no roll-outs are involved in standard RNN training. We thus consider next whether ideas coming from L2S could mitigate the limitations of MLE training for RNNs. In particular, one key property of L2S worth porting over to RNN training is that the former fully leverages structured losses information, contrarily to MLE as previously noted.

## 4 IMPROVING RNN TRAINING WITH L2S

Since we are interested in leveraging structured loss information, we can try to obtain it in the same fashion as L2S. The main tool that L2S uses in order to construct a cost-sensitive dataset is the roll-out policy. In many classical structured prediction use cases, one does not need to follow through with a policy because the "cost-to-go" that the roll-out yields is either free or easily computable from the ground truth. We are however also interested in cases where this information is unavailable, and roll-outs are needed to approximate it (e.g. for machine translation). This leads to several questions. How can we integrate roll-outs in a RNN model? How do we use this additional information, i.e. what loss do we use to train the model on? How do we make it computationally tractable?

**The SEARNN Algorithm.** The basic idea of the SEARNN algorithm is quite simple: we borrow from L2S the idea of using a *global* loss for each *local* cell of the RNN. As in L2S, we first compute a *roll-in* trajectory, following a specific roll-in policy. Then, at each step $t$ of this trajectory, we compute the costs $c_t(a)$ associated with each possible token $a$. To do so we pick $a$ at this step and

then follow a *roll-out* policy to finish the output sequence $\hat{y}_a$. We then compare $\hat{y}_a$ with the ground truth using the test error itself, rather than a surrogate. By repeating this for the $T$ steps we obtain $T$ cost vectors. We use this information to derive one *cost-sensitive* training loss for each cell, which allows us to compute an update for the parameters of the model. The full process for one cell is illustrated in Figure 1. Our losses are *global-local*, in the sense that they appear at the local level but all contain sequence-level information. Our final loss is the sum over the $T$ local losses. We provide the pseudo-code for SEARNN in Algorithm 1.

---

**Algorithm 1** SEARNN algorithm (for a simple encoder-decoder network)

---

1: Initialize the weights $\omega$ of the RNN network.
2: **for** $i$ in 1 to $N$ **do**
3:      Sample $B$ ground truth input/output structured pairs $\{(x^1, y^1), \cdots, (x^B, y^B)\}$
        # Perform the roll-in/roll-outs to get the costs. This step can be heavily parallelized.
4:      **for** $b$ in 1 to $B$ **do**
5:          Compute input features $\phi(x^b)$
           # Roll-in.
6:          Run the RNN until $t^{\text{th}}$ cell with $\phi(x^b)$ as initial state by following the roll-in policy
           (see Appendix A.2 for details in the case of reference roll-in policy)
7:          Store the sequence of hidden states in order to perform several roll-outs
8:          **for** $t$ in 1 to $T$ **do**
             # Roll-outs for all actions in order to collect the cost vector at the $t^{\text{th}}$ cell.
9:             **for** $a$ in 1 to $A$ **do**
10:               Pick a decoding method (e.g. greedy or beam search)
11:               Run the RNN from the $t^{\text{th}}$ cell to the end by first enforcing action $a$ at cell $t$,
                 and then following the decoding method.
12:               Collect the cost $c_t^b(a)$ by comparing the obtained output sequence $\hat{y}_t^b(a)$ to $y^b$
13:             **end for**
14:          **end for**
15:      **end for**
16:      Derive a loss for each cell from the collected costs
17:      Update the parameters of the network $\omega$ by doing a single gradient step
18: **end for**

---

**Choosing a multi-class classifier.** SEARNN appears quite similar to L2S, but there are a few key differences that merit more explanation. As the RNN cell can serve as a multi-class classifier, in SEARNN we could pick the cell as a (shallow) shared classifier, whose input are features extracted from the full context by the previous cells of the RNN. Instead, we pick the RNN itself, thus getting a (deep) shared classifier that also learns the features directly from the context. The difference between the two options is more thoroughly detailed in Appendix B. Arbitrarily picking a token $a$ during the roll-out phase can then be done by emulating the teacher forcing technique: if predicted tokens are fed back to the model (say if the roll-out policy requires it), we use $a$ for the next cell (instead of the prediction the cell would have output). We also use $a$ in the output sequence before computing the cost.

**Choosing a cost-sensitive loss.** We now also explain our choice for the training loss function derived from the cost vectors. One popular possibility from L2S is to go the full reduction route down to binary classification. However, this technique involves creating multiple new datasets (which is hard to implement as part of a neural network), as well as training $|\mathcal{A}|^2$ binary classifiers. Instead, we simply work with the multi-class classifier encoded by the RNN cell with training losses defined next.

We now introduce two of the more successful losses we used (although we experimented with many others, which are detailed in Appendix C.1). In the following, each loss is defined at the cell level. The global loss is the sum of all $T$ losses. $s_t(a)$ refers to the score output by cell $t$ for token $a$.

**Log-loss (LL).** A central idea in L2S is to learn the target tokens the model should aim for. This is more meaningful than blindly imposing the ground truth as target, in particular when the model has deviated from the ground truth trajectory. Golberg & Nivre (2012) refer to this technique as using *dynamic oracles*. In the context of RNN training, we call this approach *target learning*.

Our first loss is thus a simple log-loss with the minimal cost token as target:

$$\mathcal{L}_t(s_t; c_t) = -\log\left(e^{s_t(a^\star)} / \sum_{i=1}^{A} e^{s_t(i)}\right) \text{ where } a^\star = \arg\min_{a \in \mathcal{A}} c_t(a). \tag{3}$$

It is structurally similar to MLE. The only difference is that instead of maximizing the probability of the ground truth action, we maximize the probability of the best performing action with respect to the cost vector. This similarity is a significant advantage from an optimization perspective: as RNNs have mostly been trained using MLE, this allows us to leverage decades of previous work. Note that when the reference policy is to simply copy the ground truth (which is sometimes optimal, e.g. when the test error is the Hamming loss), $a^\star$ is always the ground truth token. LL with reference roll-in and roll-out is in this case *equivalent* to MLE.

**Kullback-Leibler divergence (KL).**  The log-loss approach appears to be relatively wasteful with the structured information we have access to since we are only using the minimal cost value. To exploit this information more meaningfully, we consider the following approach: we convert each cost vector into a probability distribution (e.g. through a softmax operator) and then minimize a divergence between the current model distribution $P_M$ and the "target distribution" $P_C$ derived from the costs. As the MLE objective itself can be expressed as the KL divergence between $D_{gt}$ (a Dirac distribution with full mass on the ground truth) and $P_M$, we also choose to minimize the KL divergence between $P_C$ and $P_M$. Since the costs are considered fixed with respect to the parameters of the model, our loss is equivalent to the cross-entropy between $P_C$ and $P_M$.

$$\mathcal{L}_t(s_t; c_t) = -\sum_{a=1}^{A}\left(P_C(a)\log\left(P_M(a)\right)\right) \quad \begin{aligned} &\text{where } P_C(a) = e^{-\alpha c_t(a)} / \sum_{i=1}^{A} e^{-\alpha c_t(i)} \\ &\text{and } P_M(a) = e^{s_t(a)} / \sum_{i=1}^{A} e^{s_t(i)}. \end{aligned} \tag{4}$$

$\alpha$ is a scaling parameter that controls how peaky the target distributions are. It can be chosen using a validation set. The associated gradient update discriminates between tokens based on their costs. Compared to LL, KL leverages the structured loss information more directly and thus mitigates the 0/1 nature of MLE better.

**Optimization.**  Another difference between SEARN and RNNs is that RNNs are typically trained using stochastic gradient descent, whereas SEARN is a batch method. In order to facilitate training, we decide to adapt the optimization process of LOLS, an online variant of SEARN introduced by Chang et al. (2015). At each round, we select a random mini-batch of samples, and then take a single gradient step on the parameters with the associated loss (contrary to SEARN where the reduced classifier is fully trained at each round).

Note that we do not need the test error to be differentiable, as our costs $c_t(a)$ are fixed when we minimize our training loss. This corresponds to defining a different loss at each round, which is the way it is done in L2S. In this case our gradient is unbiased. However, if instead we consider that we define a single loss for the whole procedure, then the costs depend on the parameters of the model and we effectively compute an approximation of the gradient. Whether it is possible not to fix the costs and to backpropagate through the roll-in and roll-out remains an open problem.

**Expected benefits.**  SEARNN can improve performance because of a few key properties. First, our losses leverage the test error, leading to potentially much better surrogates than MLE.

Second, all of our training losses (even plain LL) leverage the structured information that is contained in the computed costs. This is much more satisfactory than MLE which does not exploit this information and ignores nuances between good and bad candidate predictions. Indeed, our hypothesis is that the more complex the error is, the more SEARNN can improve performance.

Third, the exploration bias we find in teacher forcing can be mitigated by using a "learned" roll-in policy, which may be the best roll-in policy for L2S applications according to Chang et al. (2015).

Fourth, the loss at each cell is *global*, in the sense that the computed costs contain information about full sequences. This may help with the classical vanishing gradients problem that is prevalent in RNN training and motivated the introduction of specialized cells such as LSTMs (Hochreiter & Schmidhuber, 1997) or GRUs (Cho et al., 2014).

| Dataset | | $A$ | $T$ | Cost | **MLE** | **AC** | *roll-in* *roll-out* | **LL** learned mixed | reference learned | learned learned | **KL** learned mixed | reference learned | learned learned |
|---|---|---|---|---|---|---|---|---|---|---|---|---|---|
| OCR | | 26 | 15 | Hamming | 2.8 | – | | 1.9 | 2.5 | 1.8 | **1.0** | 1.4 | 1.1 |
| Spelling | 0.3 | 43 | 10 | edit | 19.3 | 18.7 | | **17.7** | 19.5 | 17.8 | **17.7** | 19.5 | **17.7** |
| | 0.5 | | | | 41.9 | 37.4 | | **37.1** | 43.2 | 37.6 | 38.1 | 43.2 | **37.1** |

Table 1: Comparison of the SEARNN algorithm with MLE for different cost-sensitive losses and roll-in/roll-out policies. We provide the number of actions $A$ and the maximum sequence length $T$. Note that we use 0.5 as the mixing probability for the mixed roll-out policy. We ran the ACTOR-CRITIC algorithm from Bahdanau et al. (2017) on our data splits for the spelling task and report the results in the AC column (the results reported in Bahdanau et al. (2017) were not directly comparable as they used a different random test dataset each time).

**Experiments.** In order to validate these theoretical benefits, we ran SEARNN on two datasets and compared its performance against that of MLE. For a fair comparison, we use the same optimization routine for all methods. We pick the one that performs best for the MLE baseline. Note that in all the experiments of the paper, we use greedy decoding, both for our cost computation and for evaluation. Furthermore, whenever we use a mixed roll-out we always use 0.5 as our mixin parameter, following Chang et al. (2015).

The first dataset is the optical character recognition (OCR) dataset introduced in Taskar et al. (2003). The task is to output English words given an input sequence of handwritten characters. We use an encoder-decoder model with GRU cells (Cho et al., 2014) of size 128. For all runs, we use SGD with constant step-size 0.5 and batch size of 64. The cost used in the SEARNN algorithm is the Hamming error. We report the total Hamming error, normalized by the total number of characters on the test set.

The second dataset is the Spelling dataset introduced in Bahdanau et al. (2017). The task is to recover correct text from a corrupted version. This dataset is synthetically generated from a text corpus (One Billion Word dataset): for each character, we decide with some fixed probability whether or not to replace it with a random one. The total number of tokens $A$ is 43 (alphabet size plus a few special characters) and the maximum sequence length $T$ is 10 (sentences from the corpus are clipped). We provide results for two sub-datasets generated with the following replacement probabilities: 0.3 and 0.5. For this task, we follow Bahdanau et al. (2017) and use the edit distance as our cost. It is defined as the edit distance between the predicted sequence and the ground truth sequence divided by the ground truth length. We reuse the attention-based encoder-decoder model with GRU cells of size 100 described in (Bahdanau et al., 2017). For all runs, we use the Adam optimizer (Kingma & Ba, 2015) with learning rate 0.001 and batch size of 128. Results are given in Table 1, including ACTOR-CRITIC (Bahdanau et al., 2017) runs on our data splits as an additional baseline.

**Key takeaways.** First, SEARNN outperforms MLE by a significant margin on the two different tasks and datasets, which confirms our intuition that taking structured information into account enables better performance. Second, we observed that the best performing losses were those structurally close to MLE – LL and KL – whereas others (detailed in Appendix C.1) did not improve results. This might be explained by the fact that RNN architectures and optimization techniques have been evolving for decades with MLE training in mind. Third, the best roll-in/out strategy appears to be combining a learned roll-in and a mixed roll-out, which is consistent with the claims from Chang et al. (2015). Fourth, although we expect SEARNN to make stronger improvements over MLE on hard tasks (where a simplistic roll-out policy – akin to MLE – is suboptimal), we do get improvements even when outputting the ground truth (regardless of the current trajectory) is the optimal policy.

## 5 SCALING UP SEARNN

While SEARNN does provide significant improvements on the two tasks we have tested it on, it comes with a rather heavy price, since a large number of roll-outs (i.e. forward passes) have to be run in order to compute the costs. This number, $|\mathcal{A}|T$, is proportional both to the length of the sequences, and to the number of possible tokens. SEARNN is therefore not directly applicable to tasks with large output sequences or vocabulary size (such as machine translation) where computing so many forward passes becomes a computational bottleneck. Even though forward passes can be parallelized more heavily than backward ones (because they do not require maintaining activations in memory), their asymptotic cost remains in $\mathcal{O}(dT)$, where $d$ is the number of parameters of the model.

| Dataset | | **MLE** | **LL** | **KL** | **sLL** | | | | **sKL** | | | |
|---|---|---|---|---|---|---|---|---|---|---|---|---|
| | | | | | uni. | pol. | bias. | top-k | uni. | pol. | bias. | top-k |
| OCR | | 2.8 | 1.9 | 1.0 | 1.7 | 1.8 | 1.8 | 1.5 | 1.2 | 1.2 | **0.9** | 1.4 |
| Spelling | 0.3 | 19.3 | 17.7 | 17.7 | 17.6 | 17.7 | 17.7 | **17.6** | 18.4 | 17.7 | 17.7 | 18.2 |
| | 0.5 | 41.9 | 37.1 | 38.1 | 37.0 | 37.1 | 36.6 | **36.6** | 37.8 | 37.6 | 37.1 | 38.0 |

Table 2: Comparison of the SEARNN algorithm with MLE for different datasets using the sampling approach. sLL and sKL are respectively the subsampled version of the LL and the KL losses. All experiments were run with a learned roll-in and a mixed roll-out.

There are a number of ways to mitigate this issue. In this paper, we focus on subsampling both the cells and the tokens when computing the costs. That is, instead of computing a cost vector for each cell, we only compute them for a subsample of all cells. Similarly, we also compute these costs only for a small portion of all possible tokens. The speedups we can expect from this strategy are large, since the total number of roll-outs is proportional to both the quantities we are decreasing.

**Sampling strategies.** First, we need to decide how we select the steps and tokens that we sample. We have chosen to sample steps uniformly when we do not take all of them. On the other hand, we have explored several different possibilities for token sampling. The first is indeed the uniform sampling strategy. The 3 alternative samplings we tried use the current state of our model: stochastic current policy sampling (where we use the current state of the stochastic policy to pick at random), a biased version of current policy sampling where we boost the scores of the low-probability tokens, and finally a *top-k* strategy where we take the top k tokens according to the current policy. Note that the latter strategy (*top-k*) can be seen as a simplified variant of *targeted sampling* (Goodman et al., 2016), another smarter strategy introduced to help L2S methods scale. Finally, in all strategies we always sample the ground truth action to make sure that our performance is at least as good as MLE.

**Adapting our losses to sampling.** Our losses require computing the costs of all possible tokens at a given step. One could still use LL by simply making the assumption that the token with minimum cost is always sampled. However this is a rather strong assumption and it means pushing down the scores of tokens that were not even sampled and hence could not compete with the others. To alleviate this issue, we replace the full softmax by a layer applied only on the tokens that were sampled (Jean et al., 2015). While the target can still only be in the sampled tokens, the unsampled tokens are left alone by the gradient update, at least for the first order dependency. This trick is even more needed for KL, which otherwise requires a "default" score for unsampled tokens, adding a difficult to tune hyperparameter. We refer to these new losses as sLL and sKL.

**Experiments.** The main goal of these experiments is to assess whether or not combining subsampling with the SEARNN algorithm is a viable strategy. To do so we ran the method on the same two datasets that we used in the previous section. We decided to only focus on subsampling tokens as the vocabulary size is usually the blocking factor rather than the sequence length. Thus we sampled all cells. We evaluate different sampling strategies and training losses. For all experiments, we use the learned policy for roll-in and the mixed one for roll-out and we sample 5 tokens per cell. Finally, we use the same optimization techniques than in the previous experiment.

**Key takeaways.** Results are given in Table 2. The analysis of this experiment yields interesting observations. First, and perhaps most importantly, subsampling appears to be a viable strategy to obtain a large part of the improvements of SEARNN while keeping computational costs under control. Indeed, we recover all of the improvements of the full method while only sampling a fraction of all possible tokens. Second, it appears that the best strategy for token sampling depends on the chosen loss. In the case of sLL, the *top-k* strategy performs best, whereas sKL favors the biased current policy. Third, it also seems like the best performing loss is task-dependent. Finally, this sampling technique yields a $5\times$ running time speedup, therefore validating our scaling approach.

| MLE* | Mixer* | SeaRnn (conv) | MLE† | BSO† | MLE' | AC' | MLE | SeaRnn | MLE (dropout) | SeaRnn (dropout) |
|---|---|---|---|---|---|---|---|---|---|---|
| 17.7 | 20.7 | 20.5 | 22.5 | 23.8 | 25.8 | 27.5 | 24.8 | 26.8 | 27.4 | **28.2** |

Table 3: Comparison of SeaRnn with Mixer (Ranzato et al., 2016), BSO (Wiseman & Rush, 2016) and Actor-Critic (Bahdanau et al., 2017) on the IWSLT 14 German to English machine translation dataset. The asterisk (*), dagger (†) and apostrophy (') indicate results reproduced from Ranzato et al. (2016), Wiseman & Rush (2016) and Bahdanau et al. (2017), respectively. We use a reference roll-in and a mixed roll-out for SeaRnn, along with the subsampled version of the KL loss and a scaling factor of 200. SeaRnn (conv) indicates that we used a convolutional encoder instead of a recurrent one for fair comparison with Mixer.

## 6    Neural Machine Translation.

Having introduced a cheaper alternative SeaRnn method enables us to apply it to a large-scale structured prediction task and to thus investigate whether our algorithm also improves upon MLE in more challenging real-life settings.

We choose neural machine translation as out task, and the German-English translation track of the IWSLT 2014 campaign (Cettolo et al., 2014) as our dataset, as it was used in several related papers and thus allows for easier comparisons. We reuse the pre-processing of Ranzato et al. (2016), obtaining training, validation and test datasets of roughly 153k, 7k and 7k sentence pairs respectively with vocabularies of size 22822 words for English and 32009 words for German.

For fair comparison to related methods, we use similar architectures. To compare with BSO and Actor-Critic, we use an encoder-decoder model with GRU cells of size 256, with a bidirectional encoder and single-layer RNNs. For the specific case of Mixer, we replace the recurrent encoder with a convolutional encoder as in Ranzato et al. (2016) . We use Adam as our optimizer, with an initial learning rate of $10^{-3}$ gradually decreasing to $10^{-5}$, and a batch size of 64. We select the best models on the validation set and report results both without and with dropout (0.3).

Regarding the specific settings of SeaRnn, we use a reference roll-in and a mixed roll-out. Additionally, we sample 25 tokens at each cell, following a mixed sampling strategy (detailed in Appendix C.2). We use the best performing loss on the validation set, i.e. the KL loss with scaling parameter 200.

The traditional evaluation metric for such tasks is the BLEU score (Papineni et al., 2002). As we cannot use this corpus-wide metric to compute our sentence-level intermediate costs, we adopt the alternative smoothed BLEU score of Bahdanau et al. (2017) as our cost. We use a custom reference policy (detailed in Appendix C.2). We report the corpus-wide BLEU score on the test set in Table 3.

**Key takeaways.**    First, the significant improvements SeaRnn obtains over MLE on this task (2 BLEU points without dropout) show that the algorithm can be profitably applied to large-scale, challenging structured prediction tasks at a reasonable computational cost. Second, our performance is on par or better than those of related methods with comparable baselines. Our performance using a convolutional encoder is similar to that of Mixer. Compared to BSO (Wiseman & Rush, 2016), our baseline, absolute performance and improvements are all stronger. While SeaRnn presents similar improvements to Actor-Critic, the absolute performance is slightly worse. This can be explained in part by the fact that SeaRnn requires twice less parameters during training.

Finally, the learned roll-in policy performed poorly for this specific task, so we used instead a reference roll-in. While this observation seems to go against the L2S analysis from Chang et al. (2015), it is consistent with another experiment we ran: we tried applying scheduled sampling (Bengio et al., 2015) – which uses a schedule of mixed roll-ins – on this dataset, but did not succeed to obtain any improvements, despite using a careful schedule as proposed by their authors in private communications. One potential factor is that our reference policy is not good enough to yield valuable signal when starting from a poor roll-in. Another possibility is that the underlying optimization problem becomes harder when using a learned rather than a reference roll-in.

## 7    Discussion

We now contrast SeaRnn to several related algorithms, including traditional L2S approaches (which are not adapted to RNN training), and RNN training methods inspired by L2S and RL.

**Traditional L2S approaches.** Although SEARNN is heavily inspired by SEARN, it is actually closer to LOLS (Chang et al., 2015), another L2S algorithm. As LOLS, SEARNN is a meta-algorithm where roll-in/roll-out strategies are customizable (we explored most combinations in our experiments). Our findings are in agreement with those of Chang et al. (2015): we advocate using the same combination, that is, a learned roll-in and a mixed roll-out. The one exception to this rule of thumb is when the associated reduced problem is too hard (as seems to be the case for machine translation), in which case we recommend switching to a reference roll-in.

Moreover, as noted in Section 4, SEARNN adapts the optimization process of LOLS (the one difference being that our method is stochastic rather than online): each intermediate dataset is only used for a single gradient step. This means the policy interpolation is of a different nature than in SEARN where intermediate datasets are optimized for fully and the resulting policy is mixed with the previous one.

However, despite the similarities we have just underlined, SEARNN presents significant differences from these traditional L2S algorithms. First off, and most importantly, SEARNN is a full integration of the L2S ideas to RNN training, whereas previous methods cannot be used for this purpose directly. Second, in order to achieve this adaptation we had to modify several design choices, including:

- the intermediate dataset construction, which significantly differs from traditional L2S;[3]
- the careful choice of a classifier (those used in the L2S literature do not fit RNNs well);
- the design of tailored surrogate loss functions that leverage cost information while being easy to optimize in RNNs.

**L2S-inspired approaches.** Several other papers have tried using L2S-like ideas for better RNN training, starting with Bengio et al. (2015) which introduces "scheduled sampling" to avoid the exposure bias problem. The idea is to start with teacher forcing and to gradually use more and more model predictions instead of ground truth tokens during training. This is akin to a mixed roll-in – an idea which also appears in (Daumé et al., 2009).

Wiseman & Rush (2016, BSO) adapt one of the early variants of the L2S framework: the "Learning A Search Optimization" approach of Daumé & Marcu (2005, LASO) to train RNNs. However LASO is quite different from the more modern SEARN family of algorithms that we focus on: it does not include either local classifiers or roll-outs, and has much weaker theoretical guarantees. Additionally, BSO's training loss is defined by violations in the beam-search procedure, yielding a very different algorithm from SEARNN. Furthermore, BSO requires being able to compute a meaningful loss on partial sequences, and thus does not handle general structured losses unlike SEARNN. Finally, its ad hoc surrogate objective provides very sparse sequence-level training signal, as mentioned by their authors, thus requiring warm-start.

Ballesteros et al. (2016) use a loss that is similar to LL for parsing, a specific task where cost-to-go are essentially free. This property is also a requirement for Sun et al. (2017), in which new gradient procedures are introduced to incorporate neural classifiers in the AGGREVATE (Ross & Bagnell, 2014) variant of L2S.[4] In contrast, SEARNN can be used on tasks without a free cost-to-go oracle.

**RL-inspired approaches.** In structured prediction tasks, we have access to ground truth trajectories, i.e. a lot more information than in traditional RL. One major direction of research has been to adapt RL techniques to leverage this additional information. The main idea is to try to optimize the expectation of the test error directly (under the stochastic policy parameterized by the RNN):

$$\mathcal{L}(\theta) = -\sum_{i=1}^{N} \mathbb{E}_{(y_1^i,..,y_T^i) \sim \pi(\theta)} r(y_1^i,..,y_T^i) \,. \tag{5}$$

Since we are taking an expectation over all possible structured outputs, the only term that depends on the parameters is the probability term (the tokens in the error term are fixed). This allows this

---

[3]The feature extraction is fully integrated in the model and thus learnable instead of being hand-crafted. Moreover, arbitrarily picking a token $a$ during the roll-out phase to compute the associated costs requires feeding them back to the RNN (as opposed to simply adding the decision to the context before extracting features).

[4]Sun et al. (2017)'s algorithm simply replaces the classifier in AGGREVATE with a neural network. As it is trained on an ever growing dataset, a natural gradient update is required to make the algorithm tractable.

loss function to support non-differentiable test errors, which is a key advantage. Of course, actually computing the expectation over an exponential number of possibilities is computationally intractable.

To circumvent this issue, Shen et al. (2016) subsample trajectories according to the learned policy, while Ranzato et al. (2016); Rennie et al. (2016) use the REINFORCE algorithm, which essentially approximates the expectation with a single trajectory sample. Bahdanau et al. (2017) adapt the ACTOR-CRITIC algorithm, where a second *critic* network is trained to approximate the expectation.

While all these approaches report significant improvement on various tasks, one trait they share is that they only work when initialized from a good pre-trained model. This phenomenon is often explained by the sparsity of the information contained in "sequence-level" losses. Indeed, in the case of REINFORCE, no distinction is made between the tokens that form a sequence: depending on whether the sampled trajectory is above a global baseline, all tokens are pushed up or down by the gradient update. This means good tokens are sometimes penalized and bad tokens rewarded.

In contrast, SEARNN uses "global-local" losses, with a local loss attached to each step, which contains global information since the costs are computed on full sequences. To do so, we have to "sample" more trajectories through our roll-in/roll-outs. As a result, SEARNN does not require warm-starting to achieve good experimental performance. This distinction is quite relevant, because warm-starting means initializing in a specific region of parameter space which may be hard to escape. Exploration is less constrained when starting from scratch, leading to potentially larger gains over MLE.

RL-based methods often involve optimizing additional models (baselines for REINFORCE and the critic for ACTOR-CRITIC), introducing more complexity (e.g. target networks). SEARNN does not.

Finally, while maximizing the expected reward allows the RL approaches to use gradient descent even when the test error is not differentiable, it introduces another discrepancy between training and testing. Indeed, at test time, one does not decode by sampling from the stochastic policy. Instead, one selects the "best" sequence (according to a search algorithm, e.g. greedy or beam search). SEARNN avoids this averse effect by computing costs using deterministic roll-outs – the same decoding technique as the one used at test time – so that its loss is even closer to the test loss. The associated price is that we approximate the gradient by fixing the costs, although they do depend on the parameters.

RAML (Norouzi et al., 2016) is another RL-inspired approach. Though quite different from the previous papers we have cited, it is also related to SEARNN. Here, in order to mitigate the 0/1 aspect of MLE training, the authors introduce noise in the target outputs at each iteration. The amount of random noise is determined according to the associated reward (target outputs with a lot of noise obtain lower rewards and are thus less sampled). This idea is linked to the label smoothing technique (Szegedy et al., 2016), where the target distribution at each step is the addition of a Dirac (the usual MLE target) and a uniform distribution. In this sense, when using the KL loss SEARNN can be viewed as doing *learned* label smoothing, where we compute the target distribution from the intermediate costs rather than arbitrarily adding the uniform distribution.

**Conclusion and future work.** We have described SEARNN, a novel algorithm that uses core ideas from the learning to search framework in order to alleviate the known limitations of MLE training for RNNs. By leveraging structured cost information obtained through strategic exploration, we define global-local losses. These losses provide a *global* feedback related to the structured task at hand, distributed *locally* within the cells of the RNN. This alternative procedure allows us to train RNNs from scratch and to outperform MLE on three challenging structured prediction tasks. Finally we have proposed efficient scaling techniques that allow us to apply SEARNN on structured tasks for which the output vocabulary is very large, such as neural machine translation.

The L2S literature provides several promising directions for further research. Adapting "bandit" L2S alternatives (Chang et al., 2015) would allow us to apply SEARNN to tasks where only a single trajectory may be observed at any given point (so trying every possible token is not possible). *Focused costing* (Goodman et al., 2016) – a mixed roll-out policy where a fixed number of learned steps are taken before resorting to the reference policy – could help us lift the quadratic dependency of SEARNN on the sequence length. Finally, *targeted sampling* (Goodman et al., 2016) – a smart sampling strategy that prioritizes cells where the model is uncertain of what to do – could enable more efficient exploration for large-scale tasks.

ACKNOWLEDGMENTS

We would like to thank Dzmitry Bahdanau for helping us with both the spelling and the machine translation experiments, as well as Hal Daumé for constructive feedback on both Learning to Search and an earlier version of the paper. This research was partially supported by the NSERC Discovery Grant RGPIN-2017-06936, by the ERC grant Activia (no. 307574), by a Google Research Award and by Samsung Research, Samsung Electronics.

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

# A  ALGORITHMS

## A.1  SEARN (ADAPTED FROM DAUMÉ ET AL. (2009), FIGURE 1.)

---

**Algorithm 2** SEARN algorithm

---

1: Initialize a policy $h$ with the reference policy $\pi$.
2: **for** $i$ in 1 to $N$ **do**
        # Start of round $i$.
3:     Initialize the set of cost-sensitive examples $S \leftarrow \emptyset$.
        # Create the intermediate dataset for round $i$.
4:     **for** $(x, y)$ in the ground truth input/output structured pairs **do**
          # Perform the roll-in (actually only run once).
5:       Compute predictions under the current policy, $(\hat{y}_1, ..., \hat{y}_{T_x}) \sim h, x$.
6:       **for** $t$ in 1 to $T_x$ **do**
7:         Compute input features $\phi(s_t)$ for context $s_t = (x, \hat{y}_1, ..., \hat{y}_t)$.
8:         Initialize a cost vector $c_t = \langle \rangle$.
           # Perform the roll-outs for each action to fill the cost vector.
9:         **for** each possible token $a \in \mathcal{A}$ **do**
10:           Get a full sequence $\hat{y}_t(a)$ by applying an expert policy, starting from $(x, \hat{y}_{1..t}, a)$.
11:           Collect the cost $c_t(a)$ by comparing $\hat{y}_t(a)$ and $y$.
12:         **end for**
13:         Add cost-sensitive example $(\phi, c)$ to S
14:       **end for**
15:     **end for**
16:     Learn a classifier $h'$ on $S$.
17:     Interpolate $h \leftarrow \beta h' + (1 - \beta)h$.
18: **end for**
19: Return $h$.

---

## A.2  SEARNN: REFERENCE ROLL-IN WITH AN RNN.

As mentioned in Section 3, teacher forcing can be seen as the roll-in *reference policy* of the RNN. In this section, we detail this analogy further.

Let us consider the case where we perform the roll-in up until the $t^{\text{th}}$ cell. In order to be able to perform roll-outs from that $t^{\text{th}}$ cell, a hidden state is needed. If we used a *reference policy* roll-in, this state is obtained by running the RNN until the $t^{\text{th}}$ cell by using the teacher forcing strategy, i.e. by conditioning the outputs on the ground truth. Finally, SEARNN also needs to know what the predictions for the full sequence were in order to compute the costs. When the *reference roll-in* is used, we obtain the predictions up until the $t^{\text{th}}$ cell by simply copying the ground truth. Hence, we discard the outputs of the RNN that are before the $t^{\text{th}}$ cell.

# B  DESIGN DECISIONS

**Choosing a classifier: to backpropagate or not to backpropagate?** In standard L2S, the classifier and the feature extractor are clearly delineated. The latter is a fixed hand-crafted transformation applied on the input and the partial sequence that has already been predicted. One then has to pick a classifier and its convergence properties carry over to the initial problem.

In SEARNN, we choose the RNN itself as our classifier. The fixed feature extractor is reduced to the bare minimum (e.g. one-hot encoding) and the classifier performs feature learning afterwards. In this setting, the intermediate dataset is the initial state and all previous decisions $(x, y_{1:t-1})$ combined with the cost vector.[5]

---

[5]In the encoder-decoder architecture, the decoder RNN does not receive $x$ directly, but rather $\phi(x)$, the features extracted from the input by the encoder RNN. In this case, our SEARNN classifier includes both the encoder and the decoder RNNs.

An alternative way to look at RNNs, is to consider the RNN cell as a shared classifier in its own right, and the beginning of the RNN (including the previous cells) as a feature extractor. One could then pick the RNN cell (instead of the full RNN) as the SEARNN classifier, in which case the intermediate dataset would be $(h_{t-1}, y_{t-1})$[6] (the state at the previous step, combined with the previous decision) plus the cost vector.

While this last perspective – seeing the RNN cell as the shared classifier instead of the full RNN – is perhaps more intuitive, it actually fits the L2S framework less well. Indeed, there is no clear delineation between classifier and feature extractor as these functions are carried out by different instances of the same RNN cell (and as such share weights). This means that the feature extraction in this case is learned instead of being fixed.

This choice of classifier has a direct consequence on the optimization routine. In case we pick the RNN itself, then each loss gradient has to be fully backpropagated through the network. On the other hand, if the classifier is the cell itself, then one should not backpropagate the gradient updates.

**Reference policy.** The reference policy defined by Daumé et al. (2009) picks the action which "minimizes the (corresponding) cost, assuming all future decisions are made optimally", i.e. $\arg\min_{y_t} \min_{y_{t+1:T}} l(y_{1:T}, y)$.

For the roll-in phase, this policy corresponds to always picking the ground truth, since it leads to predicting the full ground truth sequence and hence the best possible loss.

For the roll-out phase, computing this policy explicitly is easy in a few select cases. However, in the general case it is not tractable. One then has to turn to heuristics, whose performance can be relatively poor. While Chang et al. (2015) tell us that overcoming a bad reference policy can be done through a careful choice of roll-in/roll-out policies, the fact remains that the better the reference policy is, the better performance will be. Choosing this heuristic well is then quite important.

The most basic heuristic is to simply use the ground truth. Of course, one can readily see that it is not always optimal. For example, when the model skips a token and outputs the next one, $a$, instead, it may be more beneficial to also skip $a$ in the roll-out phase rather than to repeat it.

Although we mostly chose this basic heuristic in this paper, using tailored alternatives can yield better results for tasks where it is suboptimal, such as machine translation (see Appendix C.2).

## C  ADDITIONAL EXPERIMENTAL DETAILS

### C.1  LOSSES.

We now describe other losses we tried but did not perform as well (or at least not better) than the ones presented in the main text.

The first two follow the target learning principle, as LL.

**Log-loss with cost-augmented softmax (LLCAS).** LLCAS is another attempt to leverage the structured information we have access to more meaningfully, through a slight modification of LL. We add information about the full costs in the exponential, following e.g. Pletscher et al. (2010); Gimpel & Smith (2010); Hazan & Urtasun (2010).

$$\mathcal{L}_t(s_t; c_t) = -\log\left(e^{s_t(a^\star) + \alpha c_t(a^\star)} \Big/ \sum_{i=1}^{A} e^{s_t(i) + \alpha c_t(i)}\right) \text{ where } a^\star = \arg\min_{a \in \mathcal{A}} c_t(a). \quad (6)$$

$\alpha$ is a scaling parameter that ensures that the scores of the model and the costs are not too dissimilar, and can be chosen using a validation set. The associated gradient update discriminates between tokens based on their costs. Although it leverages the structured loss information more directly and thus should in principle mitigate the 0/1 nature of MLE better, we did not observe any significant improvements over LL, even after tuning the scaling parameter $\alpha$.

---

[6]One could also add $\psi(x)$, features learned from the input through e.g. an attention mechanism.

**Structured hinge loss (SHL).** The LLCAS can be seen as a smooth version of the (cost-sensitive) structured hinge loss used for structured SVMs (Tsochantaridis et al., 2005), that we also consider:

$$\mathcal{L}_t(s_t; c_t) = \max_{a \in \mathcal{A}}(s_t(a) + c_t(a)) - s_t(a^\star) \text{ where } a^\star = \arg\min_{a \in \mathcal{A}} c_t(a). \tag{7}$$

While this loss did enable the RNNs to learn, the overall performance was actually slightly worse than that of MLE. This may be due to the fact that RNNs have a harder time optimizing the resulting objective, compared to others more similar to the traditional MLE objective (which they have been tuned to train well on).

**Consistent loss.** This last loss is inspired from traditional structured prediction. Following Lee et al. (2004), we define:

$$\mathcal{L}_t(c_t) = \sum_{a \in \mathcal{A}} c_t(a) \ln(1 + \exp(\tilde{s}_t(a))) \text{ where } \tilde{s}_t(a) = s_t(a) - \frac{1}{A} \sum_{a \in \mathcal{A}} s_t(a). \tag{8}$$

Unfortunately, we encountered optimization issues and could not get significant improvements over the MLE baseline.

**KL and label smoothing.** We have seen that when the loss function is the Hamming loss, the reference policy is to simply output the ground truth. In this case, LL with a reference roll-in and roll-out is equivalent to MLE. Interestingly, in the same setup KL is also equivalent to an existing method: the label smoothing technique. Indeed, the vector of costs can be written as a vector with equal coordinates minus a one-hot vector with all its mass on the ground truth token. After transformation through a softmax operator, this yields the same target distribution as in label smoothing.

## C.2  NMT

**Custom sampling.** For this experiment, we decided to sample 15 tokens per cell according to the top-k policy (as the vocabulary size is quite big, sampling tokens with low probability is not very attractive), as well as 10 neighboring ground truth labels around the cell. The rationale for these neighboring tokens is that skipping or repeating words is quite a common mistake in NMT.

**Custom reference policy.** The very basic reference policy we have been using for the other experiments of the paper is too bad a heuristic for BLEU to perform well. Instead, we try adding every suffix in the ground truth sequence to the current predictions and we pick the one with the highest BLEU-1 score (using this strategy with BLEU-4 leads to unfortunate events when the best suffix to add is always the entire sequence, leading to uninformative costs).

**Reference roll-in.** As mentioned in Section 6, we had to switch from a learned to a reference roll-in. In addition to the existing problems of a weak reference policy (which affects a learned roll-in much more than a reference one), and the introduction of a harder optimization problem, there is another potential source of explanation: this may illustrate a gap in the standard reduction theory from the L2S framework. Indeed, the standard reduction analysis (Daumé et al., 2009; Chang et al., 2015) guarantees that the level of performance of the classifier on the reduced problem translates to overall performance on the initial problem.

However, this does not take into account the fact that the reduced problem may be harder or easier, depending on the choice of roll-in/roll-out combination. In this case, it appears that using a learned roll-in may have lead to a harder reduced problem and thus ultimately worse overall performance.

