# OpenReview forum: "SEARNN: Training RNNs with global-local losses"
_ICLR.cc/2018/Conference — Accept (Poster)_

### Official Review · AnonReviewer3 · 2017-11-27
**Fascinating and well investigated extension of L2S to RNNs**

**Rating:** 8
**Confidence:** 4

**Review:**

This paper extends the concept of global rather than local optimization from the learning to search (L2S) literature to RNNs, specifically in the formation and implementation of SEARNN. Their work takes steps to consider and resolve issues that arise from restricting optimization to only local ground truth choices, which traditionally results in label / transition bias from the teacher forced model.

The underlying issue (MLE training of RNNs) is well founded and referenced, their introduction and extension to the L2S techniques that may help resolve the issue are promising, and their experiments, both small and large, show the efficacy of their technique.

I am also glad to see the exploration of scaling SEARNN to the IWSLT'14 de-en machine translation dataset. As noted by the authors, it is a dataset that has been tackled by related papers and importantly a well scaled dataset. For SEARNN and related techniques to see widespread adoption, the scaling analysis this paper provides is a fundamental component.

This reviewer, whilst not having read all of the appendix in detail, also appreciates the additional insights provided by it, such as including losses that were attempted but did not result in appreciable gains.

Overall I believe this is a paper that tackles an important topic area and provides a novel and persuasive potential solution to many of the issues it highlights.

(extremely minor typo: "One popular possibility from L2S is go the full reduction route down to binary classification")

---

### Official Review · AnonReviewer2 · 2017-11-28
**Good ideas, but lack of comparison against previous work and unclear experiments**

**Rating:** 5
**Confidence:** 5

**Review:**

This paper proposes an adaptation of the SEARN algorithm to RNNs for generating text. In order to do so, they discuss various issues on how to scale the approach to large output vocabularies by sampling which actions the algorithm to explore.

Pros:
- Good literature review. But the future work on bandits is already happening:
Paper accepted at ACL 2017: Bandit Structured Prediction for Neural Sequence-to-Sequence Learning. Julia Kreutzer, Artem Sokolov, Stefan Riezler.


Cons:
- The key argument of the paper is that SEARNN is a better IL-inspired algorithm than the previously proposed ones. However there is no direct comparison either theoretical or empirical against them. In the examples on spelling using the dataset of Bahdanau et al. 2017, no comparison is made against their actor-critic method. Furthermore, given its simplicity, I would expect a comparison against scheduled sampling.

- A lot of important experimental details are in the appendices and they differ among experiments. For example, while mixed rollins are used in most experiments, reference rollins are used in MT, which is odd since it is a bad option theoretically. Also,  no details are given on how the mixing in the rollouts was tuned. Finally, in the NMT comparison while it is stated that similar architecture is used in order to compare fairly against previous work, this is not the case eventually, as it is acknowledged at least in the case of MIXER. I would have expected the same encoder-decoder architecture to have been used for all the methods considered.

- the two losses introduced are not really new. The log-loss is just MLE, only assuming that instead of a fixed expert that always returns the same target, we have a dynamic one. Note that the notion of dynamic expert is present in the SEARN paper too. Goldberg and Nivre just adapted it to transition-based dependency parsing. Similarly, since the KL loss is the same as XENT, why give it a new name?

- the top-k sampling method is essentially the same as the targeted exploration of Goodman et al. (2016) which the authors cite. Thus it is not a novel contribution.

- Not sure I see the difference between the stochastic nature of SEARNN and the online one of LOLS mentioned in section 7. They both could be mini-batched similarly. Also, not sure I see why SEARNN can be used on any task, in comparison to other methods. They all seem to be equally capable.

Minor comments:
- Figure 1: what is the difference between "cost-sensitive loss" and just "loss"?
- local vs sequence-level losses: the point in Ranzato et al and Wiseman & Rush is that the loss they optimizise (BLEU/ROUGE) do not decompose over the the predictions of the RNNs.
- Can't see why SEARNN can help with the vanishing gradient problem. Seem to be rather orthogonal.

---

> ### Author Response · Authors · 2017-12-15
> **Authors' response (1/3)**
>
> Reviewer2 provides an in-depth and thoughtful review. They express concerns about three potential issues: a lack of comparison to related methods, unclear experiments and erroneous novelty claims. We believe these criticisms stem for the most part from several key misunderstandings about the presented method and the claims made in the paper.
> In the following, we make explicit these misunderstandings and we strive to clarify them.
> We hope that reviewer 2 can help us improve the paper by pointing out the specific parts that they found confusing.
>
> 1. How does SeaRNN relate to other IL-inspired algorithms?
>
> "The key argument of the paper is that SEARNN is a better IL-inspired algorithm than the previously proposed ones. However there is no direct comparison either theoretical or empirical against them."
>
> We disagree with this statement and show in the following that the paper does indeed contain both theoretical and empirical comparisons, including a section (Discussion, Section 7) about theoretical comparison to related methods and a large-scale experiment where the performance of various methods is compared.
>
> First off, the main aim of the paper is to introduce a novel IL-inspired method for training RNNs which alleviates the issues associated with traditional MLE training. We then contrast different methods and explore their pros and cons. These concrete elements of comparison, both theoretical and empirical, lead us to believe that SeaRNN is indeed well-positioned.
>
> Theoretical comparisons:
> As part of this exploration, we provide numerous theoretical points of comparison in the Discussion section (Section 7):
>
> - we compare with schedule sampling (Bengio et al, 2015). They use a mixed roll-in, while we use either a reference or a learned roll-in. Furthermore, SeaRNN leverages roll-outs for estimation and custom losses, while schedule sampling simply uses the MLE loss.
> - we underline an important difference between SeaRNN and most related methods (be they RL-inspired e.g. MIXER (Ranzato et al, 2016) and Actor-Critic (Bahdanau et al, 2017) or IL-inspired e.g. BSO (Wiseman et al, 2016)): the fact that since the training signal from their loss is quite sparse, they have to use warm starting, whereas SeaRNN does not.
> - we remark that BSO requires being able to compute the evaluation metric on unfinished sequences (see the definition of the associated loss in (Wiseman and Rush, 2016, Section 4.1)). While this is technically possible for BLEU, the scores obtained this way are arguably not meaningful. In contrast, SeaRNN always computes scores on full sequences.
> - we explain that some IL-inspired methods (see Ballesteros et al, 2016 and Sun et al, 2017) require a free cost-to-go oracle, whereas SeaRNN uses roll-outs for exploration and is thus more widely applicable, albeit at a higher computational cost.
> - Incidentally, the last two points explain why we write that SeaRNN can be used on a wider amount of tasks, compared to some related methods.
>
> Empirical comparisons:
> "In the examples on spelling using the dataset of Bahdanau et al. 2017, no comparison is made against their actor-critic method. Furthermore, given its simplicity, I would expect a comparison against scheduled sampling. »
>
> As we explain in the caption of Table 1, we cannot directly compare SeaRNN to Actor-Critic on the Spelling dataset, because the authors of this paper used a random test dataset and some key hyper parameters are missing from the open source implementation (we obtained this information through private communication with them when first trying to compare our methods).
> We do provide a point of comparison with Actor-Critic (with the same architecture) on a larger scale dataset, namely IWSLT'14 de-en MT.
> Finally, we conducted thorough experiments with scheduled sampling on the NMT dataset. Unfortunately, we could not obtain any significant improvement over MLE, even with a careful schedule proposed by the authors of the scheduled sampling paper through private communication (note that no positive results on NMT were reported in the original paper either). This is reported in the main text of the paper (see Key takeaways in Section 6, at the bottom of page 8).
> If the reviewer believes this would add to the paper, we will of course run this algorithm on the OCR and Spelling datasets and report the obtained results (we have not conducted these experiments yet).
>
> All told, we believe our paper does present theoretical and empirical comparisons to related methods. We have already conducted and reported on some of the experiments the reviewer asks for.

---

> > ### Author Response · Authors · 2017-12-15
> > **Authors' response (2/3)**
> >
> > 2. Experimental details
> >
> > "A lot of important experimental details are in the appendices and they differ among experiments. »
> > "For example, while mixed rollins are used in most experiments, reference rollins are used in MT, which is odd since it is a bad option theoretically."
> > "Also, no details are given on how the mixing in the rollouts was tuned."
> > "Finally, in the NMT comparison while it is stated that similar architecture is used in order to compare fairly against previous work, this is not the case eventually, as it is acknowledged at least in the case of MIXER."
> >
> > The reviewer points out that our experimental setup is unclear. We disagree with that statement and show in the following that all of the relevant information can be found in the main text of the paper and that differences are underlined and analyzed in details. We will strive to present this information more clearly.
> >
> > First off, let us point out that there are no mixed roll-ins in any of the experiments. We compare reference and learned roll-ins for OCR and Spelling (see Table 1 and the caption of Table 2), and use reference roll-ins for NMT, as stated at the beginning of Section 6 (in the middle of page 8) and at the end of this section (see bottom of page 8).
> >
> > Second, while L2S theory indeed tells us that a learned roll-in should always be preferred to a reference one, on some datasets practitioners observe the reverse. We confirmed this with the authors of the SEARN paper (Daumé et al, 2009) through private communication.
> > We provide potential explanations in the main text of the paper (see Key takeaways in Section 6, bottom of page 8), namely:
> >
> > - either our reference policy is too weak to provide good enough training signal
> > - or the problem obtained with a learned roll-in might be harder to optimize for than its equivalent obtained with a reference roll-in -- an issue which is overlooked by classical L2S theory.
> >
> > We also explain what choice of hyper parameter we advocate, including resorting to a reference roll-in when a learned roll-in does not lead to good performance (see 'Traditional L2S approches', Section 7, top of page 9).
> > We therefore argue that this choice in hyper parameter is made explicit and is motivated in the paper.
> >
> > Third, the value of the mix-in probability for our roll-outs (0.5) is reported in the caption underneath Table 1. It is the same for all datasets. We do not report any tuning of this value because we did not perform any. We followed Chang et al (2015), where the authors indicate that their algorithm is not sensitive to this value, so we did not feel the need to optimize for it. We will add this reasoning to the paper to explain the value we took.
> >
> > Finally, we do indeed use an architecture that is different from that of MIXER. This information is reported in the main text (see Key takeaways in Section 6), as we are explicit about the architectures of related methods. The reason for this difference is that we decided to reuse the architecture used both by BSO and by Actor-Critic. We have followed their setup as closely as possible, and are not aware of any meaningful difference with our own. If our presentation is not clear enough, we are happy to add this information at any place the reviewer sees fit.
> >
> > Once again, we stress that all of this information is presented *in the main text*, and discussed at length. The only thing present in the appendix is an expanded version of the harder optimization problem hypothesis we make in 'Key takeaways' in Section 6.

---

> > > ### Author Response · Authors · 2017-12-15
> > > **Authors' response (3/3)**
> > >
> > > 3. Novelty
> > >
> > > "The two losses introduced are not really new."
> > > "The top-k sampling method is essentially the same as the targeted exploration of Goodman et al. (2016) which the authors cite. Thus it is not a novel contribution."
> > >
> > > We show that these assessments are the result of misunderstandings (in some cases we simply do not make novelty claims, and in others what we propose is actually different from the referred techniques).
> > >
> > > First, we want to reiterate the difference between a classical classification loss and a cost-sensitive loss, as these notions are fundamental to the whole field of L2S research. In a cost-sensitive classification problem, rather than having access to a single ground-truth output, one has access to a vector of costs, with one cost associated with each possible token. This unusual setup requires adapted losses. In particular, we are not aware of any other RNN training techniques which uses cost-sensitive losses, besides SeaRNN.
> > >
> > > Second, concerning the log-loss (LL), we explain that it indeed shares the structure of MLE, and replaces constant experts by dynamic ones (see ‘Log-loss’ in Section 4). We also point out that this technique is not new, even in the context of RNN training (see our reference to Ballesteros et al, (2016) in 'L2S-inspired approaches' in the Discussion section at the bottom of page 9). We do not make novelty claims in that respect.
> > > However, to our knowledge this is the first algorithm which uses the scores obtained by roll-outs to determine the value of the dynamic expert. This is the aspect of the loss which we consider to be novel.
> > > If our claim is unclear we can definitely rephrase it in a way that the reviewer deems more satisfactory.
> > >
> > > Third, we are not sure we understand the remark of the reviewer concerning the KL loss. In our setting, the KL divergence and the cross-entropy are indeed equivalent since the additional entropy term in XENT is constant with respect to the parameters of the model. We decided to call it KL as we saw this loss term as a divergence between two probability distributions (and indeed we tried several other divergences, see Appendix C).
> > > MLE can be thought of as a cross-entropy term between the model output and a Dirac distribution centered on the ground truth target.
> > > However, the difference in our setup is that we have access to a richer, non-Dirac target distribution, which we derive from the cost vectors. The novelty in our loss resides in the application of the KL divergence (or equivalently cross-entropy) in a situation where one has access to a full probabilistic distribution over the tokens in the vocabulary instead of a single target output.
> > >
> > > Finally, the top-k strategy is a simplified version of targeted sampling. Indeed, none of the strategies we test (uniform, topk, policy sampling and biased policy sampling) are novel. We acknowledge this in the main text of the paper and we make no claims about novelty with respect to these strategies.
> > >
> > > Conclusion
> > > We believe we have alleviated a number of concerns and clarified some misunderstandings which lead to unfavorable assessments about the paper. In light of these clarifications, we hope the reviewer will consider adjusting their evaluation accordingly, and helping us improve the paper through suggestions.
> > >
> > >
> > > References:
> > > Dzmitry Bahdanau, Philemon Brakel, Kelvin Xu, Anirudh Goyal, Ryan Lowe, Joelle Pineau, Aaron Courville, and Yoshua Bengio. An actor-critic algorithm for sequence prediction. In ICLR, 2017.
> > > Miguel Ballesteros, Yoav Goldberg, Chris Dyer, and Noah A Smith. Training with exploration improves a greedy stack-LSTM parser. In EMNLP, 2016.
> > > Samy Bengio, Oriol Vinyals, Navdeep Jaitly, and Noam Shazeer. Scheduled sampling for sequence prediction with recurrent neural networks. In NIPS, 2015.
> > > Kai-Wei Chang, Akshay Krishnamurthy, Alekh Agarwal, Hal Daumé, III, and John Langford. Learning to search better than your teacher. In ICML, 2015.
> > > Hal Daumé, III, John Langford, and Daniel Marcu. Search-based structured prediction. Machine Learning, 2009.
> > > Marc’Aurelio Ranzato, Sumit Chopra, Michael Auli, and Wojciech Zaremba. Sequence level training with recurrent neural networks. In ICLR, 2016.
> > > Wen Sun, Arun Venkatraman, Geoffrey J. Gordon, Byron Boots, and J. Andrew Bagnell. Deeply aggrevated: Differentiable imitation learning for sequential prediction. In ICML, 2017.
> > > Sam Wiseman and Alexander M Rush. Sequence-to-sequence learning as beam-search optimization. In EMNLP, 2016.

---

> > > > ### Comment · AnonReviewer2 · 2017-12-22
> > > > **Response to authors' response (3/3)**
> > > >
> > > > - "In particular, we are not aware of any other RNN training techniques which uses cost-sensitive losses, besides SeaRNN."
> > > >
> > > > I don't think this is the case. REINFORCE as used by Ranzato et al. (2016) for RNN training does exactly that (see eq. 11 in their paper): the difference between the reward achieved and the average expected reward is used to scale the gradient of the loss which is propagated through the network.
> > > >
> > > > - "However, to our knowledge this is the first algorithm which uses the scores obtained by roll-outs to determine the value of the dynamic expert. This is the aspect of the loss which we consider to be novel."
> > > >
> > > > A straightforward way to describe your approach is that the roll outs are used to obtain the costs, but then they are dropped and just the action has the minum action is kept. SEARN does exactly the same if one replaces the cost-sensitive learner with cost-insensitive one. The relation to Goldberg and Nivre (2012) explicitly is that they define a heuristic dynamic oracle for their task (which is very efficient to compute), while you do rollouts (which much slower, but not task specific) like SEARN and LOLS. In any case, the multiclass classification loss itself is not changed in any way, thus no new name is warranted.
> > > >
> > > > - "The novelty in our loss resides in the application of the KL divergence (or equivalently cross-entropy) in a situation where one has access to a full probabilistic distribution over the tokens in the vocabulary instead of a single target output."
> > > >
> > > > Indeed. I don't object to your application of KL-divergence/XENT, I only object to having a new name for it. Giving a new name for a loss suggests a novel loss. But as you say, no novel losses are introduced, hence no new names are warranted.
> > > >
> > > > - "Finally, the top-k strategy is a simplified version of targeted sampling. Indeed, none of the strategies we test (uniform, topk, policy sampling and biased policy sampling) are novel. We acknowledge this in the main text of the paper and we make no claims about novelty with respect to these strategies."
> > > >
> > > > Nowhere in the paper the statement "the top-k strategy is a simplified version of targeted sampling". In the section introducing it no credit is given to previous work, and it is mentioned  as a contribution of the paper in the introduction. Goodman et al is only mentioned much later in the conclusion. To avoid such misunderstandings, add this statement where you introduce the top-k strategy.
> > > >
> > > > I believe my review and comments have explicit recommendations for experiments and revisions to the text.

---

> > > > > ### Author Response · Authors · 2018-01-05
> > > > > **Specific comments (3/3)**
> > > > >
> > > > > Loss name: "But as you say, no novel losses are introduced, hence no new names are warranted."
> > > > >
> > > > > We really apologize but we still don't understand why you are saying that we use a "new name" for our cost-sensitive loss. When naming the loss "Kullback-Leibler divergence (KL)", we are simply using the standard statistic term without any intention of using a new name for the sake of sounding more novel. We simply prefer the term 'KL' to the term 'XENT' for the reasons evoked in our previous reply. We also believe that using the term ‘MLE’ instead of ‘logloss’ would be detrimental to the general understanding of the method, as the MLE training mode of RNN refers to the traditional training mode.
> > > > >
> > > > > We will be happy to revise our paper if you have an explicit recommendation on that point.
> > > > >
> > > > > Paper writing recommendation (see general answer).
> > > > >
> > > > > Thanks again for all your feedback.

---

> > > ### Comment · AnonReviewer2 · 2017-12-22
> > > **Response to authors' response (2/3)**
> > >
> > > -"First off, let us point out that there are no mixed roll-ins in any of the experiments.":
> > >
> > > Indeed, thanks for the clarification. But see my comment about the algorithmic description not really stating the option of a reference rollin policy. In any case, the mixed rollins in their extreme settings cover both reference and learned.
> > >
> > > - "Second, while L2S theory indeed tells us that a learned roll-in should always be preferred to a reference one, on some datasets practitioners observe the reverse"
> > >
> > > Indeed. If the paper was about an algorithm for a particular dataset/task, that would be OK. But SEARNN is claimed to be widely applicable, thus I expect it to be consistently defined across tasks when compared to previous work, but this is not the case.
> > >
> > > - "Third, the value of the mix-in probability for our roll-outs (0.5) is reported in the caption underneath Table 1. It is the same for all datasets.":
> > >
> > > Thanks for the clarification, couldn't have known that this is the case for all experiments.
> > >
> > > - "Finally, we do indeed use an architecture that is different from that of MIXER. This information is reported in the main text (see Key takeaways in Section 6)":
> > >
> > > Yes, but earlier it reads: "For fair comparison to related methods, we use a similar architecture"
> > >
> > > Replacing an RNN with a CNN is not similar in my opinion. As I wrote in the first part of my response, Bahdanau et al. (2017) run different experiments  for this reason.

---

> > > > ### Author Response · Authors · 2018-01-05
> > > > **Specific comments (2/3)**
> > > >
> > > > Generality of SEARNN: "But SEARNN is claimed to be widely applicable, thus I expect it to be consistently defined across tasks when compared to previous work"
> > > >
> > > > You are also concerned about the lack of generality of SEARNN due to the fact that the best rollin strategies are not always consistent within tasks. We believe this is not a problem as we consider SEARNN (as LOLS and SEARN) to be a meta algorithm, and the choice of rollin and rollout strategies to be hyperparameters of the method (similar to the mixing parameter in SEARN). We hope that this view addresses your concern.
> > > >
> > > > About the mixin rollout parameter: "couldn't have known that this is the case for all experiments."
> > > >
> > > > We have added that more explicitly (page 6, experiments paragraph). Sorry for the confusion.
> > > >
> > > > "Similar architectures": see general answer.

---

> > ### Comment · AnonReviewer2 · 2017-12-22
> > **Response to authors' response (1/3)**
> >
> > I appreciate the long response to my review. Here are some comments to the response:
> >
> > - "Theoretical comparisons:
> > As part of this exploration, we provide numerous theoretical points of comparison in the Discussion section (Section 7): ...":
> >
> > I guess I wasn't clear in what I meant by theoretical comparisons. For an example for what I meant and think necessary, see section 3 in the paper by Chang et al. 2015 (cited in the paper). Such an analysis is not conducted in the paper.
> >
> > Besides that, the concluding point: "we write that SeaRNN can be used on a wider amount of tasks, compared to some related methods." On page 9 it reads: "In contrast, S EA R NN can be used on any task." which is a much stronger claim, and is not supported. You should be clear in the paper: which methods and which (kinds of) tasks.
> >
> > - "we compare with schedule sampling (Bengio et al, 2015). They use a mixed roll-in, while we use either a reference or a learned roll-in.":
> >
> > I don't think this is correct; mixed roll-ins depending on the parameterization span the spectrum from reference to learned, and everything in between.
> >
> > - "As we explain in the caption of Table 1, we cannot directly compare SeaRNN to Actor-Critic on the Spelling dataset, because the authors of this paper used a random test dataset and some key hyper parameters are missing from the open source implementation (we obtained this information through private communication with them when first trying to compare our methods)."
> >
> > In this case you should run the open-source implementation on your data splits to obtain comparable results to yours.
> >
> > - "We do provide a point of comparison with Actor-Critic (with the same architecture) on a larger scale dataset, namely IWSLT'14 de-en MT."
> >
> > In the text of the paper it reads "For fair comparison to related methods, we use a similar architecture". In any case, you cannot be similar to both the RNN encoder of Bahdanau et al. (2017) and Wiseman and Rush (2016) and the CNN encoder of Ranzato et al. (2016) unless you try both, which is in fact what Bahdanau et al. (2017) did. I expect you to do the same here.
> >
> > - "Finally, we conducted thorough experiments with scheduled sampling on the NMT dataset. Unfortunately, we could not obtain any significant improvement over MLE..."
> >
> > Indeed, apologies for having missed this point; I was looking for it in the OCR experiments section. However, the explanation given is convincing: "Another possibility is that the underlying optimization problem becomes harder when using a learned rather than a reference roll-in." If this is the case, then it should have been a problem for SEARNN which obtains its best results with learned roll-ins on OCR and spelling?
> >
> > And related point: you specify the algorithm in the appendix saying:
> > "Run the RNN until t th cell with φ(x b ) as initial state by following the roll-in policy"
> > Using the RNN means learned, or at least mixed, but  cannot be reference which wouldn't be using the RNN. It is confusing that your only experimental comparison with other methods doesn't use the rollins stated in the algorithmic description.

---

> > > ### Author Response · Authors · 2018-01-05
> > > **Specific comments (1/3)**
> > >
> > > Theoretical comparisons: We also agree that having theoretical results such as the one presented in Chang's paper would be a nice addition to our paper. We leave this interesting developments as future work.
> > >
> > > Experimental comparison & paper writing: see general answer.
> > >
> > > Hypothesis: Concerning our hypothesis "Another possibility is that the underlying optimization problem becomes harder when using a learned rather than a reference roll-in.", we want to stress again that we don't claim this is what happens but we are making what we think is a plausible conjecture. The optimization problem changes with the task and hence we don't see why our point is not valid even if we don't observe the same behavior for two different tasks.
> > >
> > > Algorithm: see general answer.

---

### Official Review · AnonReviewer1 · 2017-12-06
**Successful application of L2S to RNN training**

**Rating:** 7
**Confidence:** 3

**Review:**

The paper proposes new RNN training method based on the SEARN learning to search (L2S) algorithm and named as SeaRnn. It proposes a way of overcoming the limitation of local optimization trough the exploitation of the structured losses by L2S. It can consider different classifiers and loss functions, and a sampling strategy for making the optimization problem scalable is proposed. SeaRnn improves the results obtained by MLE training in three different problems, including a large-vocabulary machine translation. In summary, a very nice paper.

Quality: SeaRnn is a well rooted and successful application of the L2S strategy to the RNN training that combines at the same time global optimization and scalable complexity.

Clarity: The paper is well structured and written, with a nice and well-founded literature review.

Originality: the paper presents a new algorithm for training RNN based on the L2S methodology, and it has been proven to be competitive in both toy and real-world problems.

Significance: although the application of L2S to RNN training is not new, the contribution to the overcoming the limitations due to error propagation and MLE training of RNN is substantial.

---

### Author Response · Authors · 2017-12-15
**Authors' response**

We thank the reviewers for their thorough and detailed evaluations. We are grateful for all the positive feedback given by the reviewers and their suggestions.
Reviewer 2 expresses some concerns about the paper which seem due to several misunderstandings; we clarify these in a specific response.

---

### Author Response · Authors · 2018-01-05
**General answer to Reviewer 2**

To begin with, we would like to express our gratitude for your very detailed comments to our response, which helped us understand the points in your initial review a lot better.

We have proceeded to a revision of the paper according to your comments in order to clarify and correct some of our claims. First, we are now more specific on the kind of tasks SEARNN can tackle (see notably page 9 last sentence). Second, we have added a specific comment about the difference in architecture with MIXER, and made explicit that a direct comparison is not meaningful (see footnote 2 page 8).  Third, we agree that there is a subtlety about the algorithm that can lead to a misunderstanding, especially in the context of reference roll-in. In order to improve the clarity of this point, we have been more explicit in the main paper (see Links to RNNs paragraph, section 3) and have added a description of how exactly a reference roll-in is achieved with an RNN (section A.3). The main point is that even for the reference roll-in strategy one still need to use the RNN in order to obtain the hidden states that will be used to initialize the roll-outs. The only difference is that the ground truth is passed to the next cell instead of the model's prediction (teacher forcing like). Finally, following your legitimate recommendation concerning the top-k strategy, we have added the citation to Goodman et al. and our statement at the moment we introduce it (Sampling strategies, page 7).

Finally, we agree with you that the two additional experiments that you are requesting, namely running the actor critic method on our spelling dataset and running SEARNN on the MIXER architecture would be valuable additions to the paper. We will include them in a future revision as soon as we obtain results, but unfortunately we haven’t yet had time to finish these experiments due to the holiday period and the length of training. We apologize for these setbacks.
However, although they would add to the quality of the paper, we still believe that in its current form the paper already contains enough material to deserve publication.

Thank you again for your valuable feedback, which enables us to improve the quality of our paper. Note that we have answered other more specific points below your answers.

---

> ### Comment · AnonReviewer2 · 2018-01-12
> **Response to general answer**
>
> While the paper has been improved, my main concern "lack of comparison against previous work and unclear experiments" remains. As the authors acknowedge, the experiments I have argued are missing are sensible and they would provide the evidence to support the claims about the suitability of the proposed IL-based method to RNN training. However they are not there, and thus, while the idea is good, I don't believe it is ready for publication and hence I stand by my original rating. Also, I still believe that a paper introducing a new algorithm doesn't help itself by putting the algorithm in the appendix. Also note, that previous work like SEARN and LOLS is explicit about the choices of rollins and rollouts, they are not "hyper-parameters".

---

### Decision · Program_Chairs · 2018-01-29
**ICLR 2018 Conference Acceptance Decision**

**Decision:**

Accept (Poster)

**Comment:**

This paper generally presents a nice idea, and some of the modifications to searn/lols that the authors had to make to work with neural networks are possibly useful to others. Some weaknesses exist in the evaluation that everyone seems to agree on, but disagree about importance (in particular, comparison to things like BLS and Mixer on problems other than MT).

A few side-comments (not really part of meta-review, but included here anyway):
- Treating rollin/out as a hyperparameter is not unique to this paper; this was also done by Chang et al., NIPS 2016, "A credit assignment compiler..."
- One big question that goes unanswered in this paper is "why does learned rollin (or mixed rollin) not work in the MT setting." If the authors could add anything to explain this, it would be very helpful!
- Goldberg & Nivre didn't really introduce the _idea_ of dynamic oracles, they simply gave it that name (e.g., in the original Searn paper, and in most of the imitation learning literature, what G&n call a "dynamic oracle" everyone else just calls an "oracle" or "expert")